# Reflux Esophagitis and Fatigue: Are They Related?

**DOI:** 10.3390/jcm10081588

**Published:** 2021-04-09

**Authors:** Sung-Goo Kang, Hyun jee Hwang, Youngwoo Kim, Junseak Lee, Jung Hwan Oh, Jinsu Kim, Chul-Hyun Lim, Seung Bae Youn, Sung Hoon Jung

**Affiliations:** 1Department of Family Medicine, St. Vincent Hospital, College of Medicine, The Catholic University of Korea, Suwon 16247, Korea; hippo94@naver.com; 2Department of Internal Medicine, Eunpyeong St. Mary’s Hospital, College of Medicine, The Catholic University of Korea, Seoul 03312, Korea; hhj1107@naver.com (H.j.H.); cureisyou@gmail.com (Y.K.); saintmary03@naver.com (J.L.); ojh@catholic.ac.kr (J.H.O.); jinsu23@naver.com (J.K.); diluck@catholic.ac.kr (C.-H.L.); sbyoon@catholic.ac.kr (S.B.Y.)

**Keywords:** gastroesophageal reflux disease, reflux esophagitis, esophagus, fatigue, depression

## Abstract

Gastroesophageal reflux disease (GERD) is a chronic, recurrent disease. Reflux esophagitis can interfere with sleep via acid reflux, which can cause daytime sleepiness or fatigue. However, little is known about the association between reflux esophagitis and fatigue. Objectives: We evaluated the association between fatigue and reflux esophagitis in subjects seen at health check-ups. Methods: Consecutive patients who were scheduled for screening endoscopies were enrolled prospectively at the Comprehensive Medical Examination Center of St. Vincent Hospital and Eunpyeong St. Mary’s Hospital, Korea. Three validated questionnaires were used to assess fatigue, daytime hypersomnolence, anxiety, and depression: the Multidimensional Fatigue Inventory—Korean version (MFI-K), Epworth Sleepiness Scale (ESS), and Hospital Anxiety and Depression Scale (HADS). Results: We investigated 497 consecutive eligible subjects. The reflux esophagitis and symptomatic GERD groups comprised 103 (20.7%) and 92 (18.5%) subjects, respectively. The MFI-K total, ESS, HADS-anxiety, and HADS-depression scores did not differ between the esophagitis and non-esophagitis groups (50.0 ± 11.5 vs. 49.7 ± 10.9, *p* = 0.661; 6.2 ± 2.8 vs. 6.1 ± 3.1, *p* = 0.987; 5.8 ± 3.1 vs. 5.2 ± 3.2, *p* = 0.060; 6.2 ± 3.6 vs. 6.0 ± 3.3, *p* = 0.561). However, the MFI-K total, ESS, HADS-anxiety, and HADS-depression scores were higher in the symptomatic group than in the non-symptomatic group (54.7 ± 12.7 vs. 48.6 ± 10.3, *p *< 0.001; 7.1 ± 3.5 vs. 5.9 ± 2.9, *p* = 0.002; 6.4 ± 3.3 vs. 5.1 ± 3.1, *p* < 0.001; 7.5 ± 4.0 vs. 5.7 ± 3.1, *p *< 0.001). Multiple regression analysis showed that the MFI-K total was correlated with GERD symptoms (*p* = 0.021), women (*p* = 0.001), anxiety (*p* < 0.001), and depression (*p *< 0.001). Conclusion: There was no statistically significant association in which reflux esophagitis could cause daytime sleepiness, fatigue, anxiety, or depression. However, fatigue was associated with GERD symptoms, women, anxiety, and depression. Further studies should clarify the association between fatigue and reflux esophagitis.

## 1. Background

Gastroesophageal reflux disease (GERD) is a chronic, recurrent disease with a prevalence of 20% in western countries, and disease-related symptoms are common in these populations [1,2]. Recently, the prevalence of GERD has increased significantly in Korea, and symptomatic GERD is also increasing [3]. Although a few studies have identified asymptomatic reflux esophagitis in some patients, most patients report regurgitation and heartburn [4,5,6], and atypical symptoms can include various other esophageal and extra-esophageal symptoms [7]. Reflux symptoms can also cause sleep disturbances or emotional dysfunction, while anxiety and depressive symptoms can conversely be associated with reflux symptoms [8,9].

Fatigue is a very subjective symptom and can be caused by a variety of organic disorders, including malignant diseases, and by psychological disorders, such as depression [10,11,12]. Sleep disturbances can also cause fatigue, and reflux esophagitis is likely to cause sleep disturbances because of acid regurgitation [13,14]. To our knowledge, however, no study has directly examined the relationship between fatigue and reflux esophagitis. Recently, the author verified the reliability and validity of the Multidimensional Fatigue Inventory—Korean version (MFI-K) as a multidimensional instrument for assessing fatigue in the Korean population [15]. MFI-K research has shown the possibility that gender and age can affect fatigue in Koreans.

Therefore, this study evaluated the relative impact of GERD on fatigue compared to other variables such as age, sex, and psychological disorders, all of which could cause fatigue. We also investigated whether endoscopy-proven esophagitis or GERD symptoms were associated with fatigue.

## 2. Methods

### 2.1. Patients and Study Design

This prospective survey study was conducted at the Comprehensive Medical Examination Centers of St. Vincent Hospital and Eunpyeong St. Mary’s Hospital, Korea. The study was approved by the Institutional Review Board of The Catholic University of Korea (PC19QESI0025) on 16 April 2019. Written informed consent was obtained from all subjects. The study protocol conformed to the ethical guidelines of the 1975 Declaration of Helsinki, as reflected in a prior approval by the institution’s human research committee.

Consecutive patients who were scheduled for screening endoscopies at the Comprehensive Medical Examination Centers of both hospitals were enrolled prospectively. We excluded patients who had severe comorbidities and took medications that could act on the central nervous system through a medical questionnaire. We also excluded patients who had advanced stomach cancer, peptic ulcer disease (active gastric or duodenal ulcer), anemia (male < 14 g/dL, female < 12 g/dL), or positive stool occult blood according to test results, including endoscopy.

All patients underwent gastro-duodenoscopy by well-trained endoscopists, and the findings were reviewed by expert endoscopists. The presence and extent of reflux esophagitis was classified using the Los Angeles (LA) classification. Minimal change in reflux esophagitis was excluded. The presence of GERD symptoms was distinguished by the presence of regurgitation and heartburn.

### 2.2. Questionnaires

Three validated questionnaires were administered to assess fatigue, daytime hypersomnolence, anxiety, and depression.

### 2.3. Multidimensional Fatigue Inventory—Korean Version (MFI-K)

Useful tools for assessing fatigue are either one- or multidimensional instruments. The most commonly used one-dimensional instruments for fatigue are the Visual Analogue Scale (VAS) and the Fatigue Severity Scale (FSS) [16,17]. One of the most useful multidimensional scaling tools for fatigue research is the MFI, which has five subscales: general fatigue, physical fatigue, reduced activity, reduced motivation, and mental fatigue [18]. The MFI-K has proven validity and reliability [15]. We used the MFI-K to assess fatigue in this study.

### 2.4. Epworth Sleepiness Scale (ESS)

The ESS was used to assess daytime sleepiness. This is a self-assessment tool consisting of eight questions, each of which is scored on a Likert scale from 0 to 3 points. The higher the ESS score, the worse the daytime sleepiness; a score above 10 indicates excessive daytime sleepiness [19].

### 2.5. Hospital Anxiety and Depression Scale (HADS)

The HADS was developed to measure the anxiety and depression of patients who visit a hospital for a short time; the Korean version of HADS has proven validity and reliability [20]. The HADS consists of 14 questions, with seven odd-numbered ones assessing anxiety and seven even-numbered ones for depression. Each is scored on a 4-point scale (0–3 points). The higher the score, the greater the anxiety and depression. We used the HADS to assess anxiety and depression in this study.

### 2.6. Statistical Analyses

The categorical variables in each group are presented as numbers with percentages and were compared using the chi-square test. The continuous variables in each of the two groups are presented as means ± standard deviations and were compared using the Analysis of covariance (ANCOVA) after adjusting for age and gender. Power for the comparisons was calculated by G-Power 3.1. The correlations of MFI-K with ESS and HADS were tested using Pearson’s correlation. In addition, we performed a multiple regression analysis with the fatigue score as a dependent variable and GERD (yes or no), age, gender, and HADS scores as independent variables. *p*-values less than 0.05 were defined as statistically significant. All statistical analyses were performed using the Statistical Package for the Social Sciences (SPSS) ver. 18.0 (SPSS, Chicago, IL, USA).

## 3. Results

### 3.1. Baseline Characteristics

The study initially enrolled 500 subjects. Three were excluded and the remaining 497 eligible cases were analyzed. Their mean age was 50.52 ± 10.13 years, and 308 (62%) were males. The reflux esophagitis and symptomatic GERD groups comprised 103 (20.7%) and 92 (18.5%) subjects, respectively. Table 1 lists the baseline characteristics of each group. Eighty-eight patients (17.7%) had hypertension, 44 (8.9%) had type 2 diabetes, and 62 (12.5%) had dyslipidemia. The numbers of LA-A, -B, -C, and -D were 52, 38, 12, and 1, respectively.

### 3.2. Questionnaires: MFI-K, ESS, and HADS

The mean MFI-K total score did not differ between the esophagitis and non-esophagitis groups (50.0 ± 11.5 vs. 49.7 ± 10.9, *p* = 0.661, Power for comparisons = 0.057), although it was higher in the symptomatic group than in the non-symptomatic group (54.7 ± 12.7 vs. 48.6 ± 10.3, *p *< 0.001, Power for comparisons = 0.995). There were no differences in scores with or without esophagitis for each MFI-K subscale. However, all subscales scored higher in the symptomatic group (Table 2). The mean ESS score did not differ between the esophagitis and non-esophagitis groups (6.2 ± 2.8 vs. 6.1 ± 3.1, *p* = 0.987, Power for comparisons = 0.061), while the mean ESS score was higher in the symptomatic group than in the non-symptomatic group (7.1 ± 3.5 vs. 5.9 ± 2.9, *p* = 0.002, Power for comparisons = 0.894). In the HADS, there were no differences in scores between groups with and without esophagitis, but the anxiety and depression scores were higher in the symptomatic group (6.4 ± 3.3 vs. 5.1 ± 3.1, *p *< 0.001, Power for comparisons = 0.937; 7.5 ± 4.0 vs. 5.7 ± 3.1, Power for comparisons = 0.991).

Table 3 shows that the endoscopy-negative reflux disease (ENRD) group had higher MFI-K, ESS, and HADS scores, especially the MFI-K total score and HADS depression subscale (53.6 ± 11.9 vs. 48.2 ± 9.8, *p* = 0.002, Power for comparisons = 0.880; 7.2 ± 3.8 vs. 5.7 ± 3.1, *p* = 0.008, Power for comparisons = 0.785) compared with the asymptomatic erosive esophagitis group. Regardless of the presence of erosive esophagitis, MFI-K, ESS, and HADS were correlated (*p *< 0.001) (Table 4). Multiple regression analysis showed that MFI-K total was correlated with women (*p* = 0.001), GERD symptoms (*p* = 0.021), anxiety (*p* < 0.001), and depression (*p* < 0.001) (Table 5).

## 4. Discussion

Fatigue is a vague, subjective symptom that is one of the most common encountered in primary care. Fatigue can accompany many different diseases, which range from organic to psychological, and can affect quality of life [11,12,21]. As in western countries, GERD is increasing in Asia and Korea [22,23]. Theoretically, GERD can cause fatigue and daytime sleepiness via sleep disturbance due to acid regurgitation. We examined whether there is a connection between GERD and fatigue.

The study results showed that fatigue and erosive esophagitis were not correlated with the MFI-K total score or with the general or physical fatigue, mental fatigue, reduced activity, or motivation subscales. Daytime sleepiness was not associated with erosive esophagitis. These findings are inconsistent with epidemiological studies that report that people who experience nighttime heartburn have sleep disorders that alter daytime performance [14,24,25,26,27]. One systematic review found that GERD was associated with sleep disturbance, and this association appeared to be bidirectional [13]. Although not all sleep disturbances due to acid regurgitation cause daytime sleepiness, nighttime sleep disturbance could cause daytime sleepiness. However, the ESS score was not higher in patients with endoscopy-proven erosive esophagitis in this study. This suggests that acid contact and regurgitation alone cannot cause daytime sleepiness and that a variety of other factors, including psychological causes, cause daytime sleepiness. Similarly, fatigue may also represent a psychophysiological symptom complex. A few studies have found relations between reflux esophagitis and fatigue, while just one study found no correlation of fatigue with active reflux esophagitis or peptic ulcer [28]. Those authors emphasized that psychosocial stress affects reflux esophagitis and is correlated with the severity of reflux esophagitis. This earlier study used the FSS, a one-dimensional instrument of fatigue. We used the MFI-K, a more complex, multidimensional measure of fatigue, to find a more detailed link between fatigue and erosive esophagitis, but we found no correlation.

Fatigue and depression are highly correlated. Symptoms and prevalence of depression increase significantly in individuals with high fatigue. In particular, the association between fatigue and depression is completely independent of the overlapping symptoms. While many factors play a role in the development of depression and fatigue, both are associated with increased activation of inflammation in the immune system [29,30]. Most of the GERD studies involving anxiety and depression focused on patient symptoms, and, in almost all studies, anxiety and depression seemed to affect GERD symptoms bidirectionally [8,31,32,33]. In our subgroup analysis of GERD symptoms, depression and anxiety were strongly associated with symptoms, similarly to other studies. In contrast to the comparison of the groups with or without endoscopy-proven erosive esophagitis, GERD symptoms such as heartburn and regurgitation were correlated with fatigue in the MFI-K total and subscale scores. The ESS score was also much higher in the GERD symptom group. These findings indicate that the presence of symptoms is related more to fatigue and daytime sleepiness than to endoscopic findings.

GERD can be classified into erosive (ERD), endoscopy-negative reflux disease (ENRD), and asymptomatic erosive esophagitis. Asymptomatic erosive esophagitis has a reported prevalence of 20~45% [4,5,6,34]. The overall prevalence of ENRD and asymptomatic erosive esophagitis was 15.3% and 17.5%, respectively. Interestingly, 84.5% of those with ERD had no symptoms, probably because only typical symptoms such as heartburn or degeneration are considered GERD symptoms and most patients had relatively mild erosive esophagitis, such as LA grades A and B. In the present study, fatigue and depression were common in the ENRD group. Daytime sleepiness also tended to increase in the ENRD group, but the difference was not significant. These findings suggest that organic causes, visceral sensitivity, esophageal motility, and psychological factors can affect ENRD, as proposed in other studies [35,36]. Furthermore, fatigue was correlated with depression, anxiety, and daytime sleepiness, regardless of the presence of erosive esophagitis. In mild erosive esophagitis, fatigue and daytime sleepiness can be caused by psychological factors rather than organic factors, such as acid regurgitation.

Although some studies have reported that fatigue is not related to age, in general, fatigue is more common in adults than in children and adolescents, and it has been reported that the frequency of fatigue increases from middle age to old age [37]. In this study, age was not associated with fatigue, which may be due to the relatively small number of elderly subjects. Few studies have analyzed gender and fatigue in Koreans. In some meta-analysis studies, women showed a prevalence of chronic fatigue syndrome approximately 1.5 to 3 times higher than that in men. This study also found that women had a higher degree of fatigue than men [38,39].

This study had several limitations. First, we only surveyed the presence of reflux symptoms and did not assess reflux symptom burden. The degree of reflux symptoms could affect the degree of fatigue, depression, or anxiety. Second, only a relatively small number of the study participants with erosive esophagitis had LA class C (*n* = 12) or D (*n* = 1) reflux esophagitis. Therefore, it is necessary to investigate relationships between fatigue, anxiety, and depression in patients with LA-C or LA-D reflux esophagitis. Third, few variables related to socioeconomic status, such as household income and marital status, which may affect patient fatigue, depression, and anxiety, were included. Finally, diet has a significant impact on reflux symptoms [40], but we did not survey the diets of the subjects in this study.

Nevertheless, this study has strengths. First, it is the first study to use the MFI-K, a multidimensional scaling tool for fatigue validated by the author, to examine the relationship between GERD and fatigue. Second, all subjects underwent endoscopy by well-trained endoscopists and the findings were reviewed by expert endoscopists.

## 5. Conclusions

Fatigue was associated with GERD symptoms, women, anxiety, and depression, but there was no statistically significant association in which reflux esophagitis could cause daytime sleepiness, fatigue, anxiety, or depression. Therefore, psychological factors are likely to cause fatigue and daytime sleepiness, rather than organic factors related to acid regurgitation. Further studies should clarify the association between fatigue and reflux esophagitis.

## Figures and Tables

**Table 1 jcm-10-01588-t001:** Baseline characteristics of the study population.

	Reflux Esophagitis No	Reflux Esophagitis Yes	*p-*Value	GERD Symptoms No	GERD Symptoms Yes	*p*-Value
Age (years)	50.5 ± 10.0	50.7 ± 10.6	0.809	50.5 ± 10.2	50.4 ± 9.8	0.936
Sex			0.111			0.721
Men	237 (60.2)	71 (68.9)		249 (61.5)	59 (64.1)	
Women	157 (39.8)	32 (31.1)		156 (38.5)	33 (35.9)	
BMI (kg/m^2^)	23.3 ± 6.1	22.0 ± 8.6	0.160	23.2 ± 6.7	22.3 ± 6.9	0.242
WC (cm)	85.1 ± 9.2	85.8 ± 10.2	0.567	85.5 ± 9.3	84.4 ± 9.8	0.374
Smoking			0.859			0.002
Non-smoker	188 (47.7)	46 (44.7)		193 (47.7)	41 (44.6)	
Ex-smoker	108 (27.4)	28 (27.2)		112 (27.7)	24 (26.1)	
Current smoker	59 (15.0)	16 (15.5)		51 (12.6)	24 (26.1)	
No answer	39 (9.9)	13 (12.6)		49 (12.1)	3 (3.3)	
Medical History						
Hypertension	66 (16.8)	22 (21.4)	0.310	71 (17.6)	17 (18.5)	0.880
Type 2 diabetes	31 (7.9)	13 (12.6)	0.171	39 (9.7)	5 (5.5)	0.306
Dyslipidemia	50 (12.7)	12 (11.7)	0.868	48 (11.9)	14 (15.2)	0.384
Alcohol			0.392			0.902
<1 time/week	133 (37.5)	29 (32.2)		129 (36.2)	33 (37.1)	
≥1 time/week	222 (62.5)	61 (67.8)		227 (63.8)	56 (62.9)	

Data are mean ± SD or *N* (%). *p*-values were obtained by independent T-tests or chi-square tests. Abbreviations: GERD, gastroesophageal reflux disease; BMI, body mass index; WC, waist circumference.

**Table 2 jcm-10-01588-t002:** Multidimensional Fatigue Inventory—Korean version (MFI-K) and reflux esophagitis.

	Reflux Esophagitis No	Reflux Esophagitis Yes	*p*-Value	Power for Comparisons	GERD Symptoms No	GERD Symptoms Yes	*p*-Value	Power for Comparisons
MFI-K								
General and Physical Fatigue	17.1 ± 4.6	17.3 ± 5.0	0.534	0.066	16.7 ± 4.5	19.1 ± 5.2	<0.001	0.989
Mental Fatigue	13.7 ± 3.7	13.7 ± 3.9	0.864	0.050	13.4 ± 3.5	15.2 ± 4.1	<0.001	0.982
Reduced Activity	8.7 ± 2.6	8.7 ± 3.0	0.814	0.050	8.5 ± 2.7	9.4 ± 2.9	0.003	0.788
Motivation	10.2 ± 3.1	10.3 ± 2.7	0.755	0.061	10.0 ± 2.9	11.0 ± 3.4	0007	0.776
MFI-K total score	49.7 ± 10.9	50.0 ± 11.5	0.661	0.057	48.6 ± 10.3	54.7 ± 12.7	<0.001	0.995
ESS	6.1 ± 3.1	6.2 ± 2.8	0.987	0.061	5.9 ± 2.9	7.1 ± 3.5	0.002	0.894
HADS								
Anxiety	5.2 ± 3.2	5.8 ± 3.1	0.060	0.401	5.1 ± 3.1	6.4 ± 3.3	<0.001	0.937
Depression	6.0 ± 3.3	6.2 ± 3.6	0.561	0.081	5.7 ± 3.1	7.5 ± 4.0	<0.001	0.991

Data are means ± SD. *p*-values were obtained by ANCOVA after adjusting for age and gender. Power for the comparisons was calculated by G-Power 3.1 Abbreviations: MFI-K, Multidimensional Fatigue Inventory—Korean version; ESS, Epworth Sleepiness Scale; HADS, Hospital Anxiety and Depression Scale; ANCOVA, Analysis of covariance.

**Table 3 jcm-10-01588-t003:** Comparison of asymptomatic erosive esophagitis group and endoscopy-negative reflux disease (ENRD) group.

	Asymptomatic Erosive Esophagitis (*n* = 87)	ENRD (*n* = 76)	*p*-Value	Power for Comparisons
MFI-K				
General and Physical Fatigue	16.5 ± 4.4	18.4 ± 4.9	0.010	0.733
Mental Fatigue	13.3 ± 3.4	15.0 ± 3.8	0.002	0.847
Reduced Activity	8.5 ± 2.9	9.3 ± 2.8	0.078	0.427
Motivation	10.0 ± 2.4	10.8 ± 3.4	0.080	0.406
MFI-K Total Score	48.2 ± 9.8	53.6 ± 11.9	0.002	0.880
ESS	6.0 ± 2.8	7.0 ± 3.6	0.052	0.501
HADS				
Anxiety	5.4 ± 2.8	6.1 ± 3.1	0.152	0.323
Depression	5.7 ± 3.1	7.2 ± 3.8	0.008	0.785

Data are means ± SD. *p*-values were obtained by ANCOVA after adjusting for age and gender. Power for the comparisons was calculated by G-Power 3.1. Abbreviations: eRE, erosive reflux esophagitis; GERD, gastroesophageal reflux disease; ENRD, endoscopy-negative reflux disease; MFI-K, Multidimensional Fatigue Inventory—Korean version; ESS, Epworth Sleepiness Scale; HADS, Hospital Anxiety and Depression Scale.

**Table 4 jcm-10-01588-t004:** Correlation between MFI-K and ESS and HADS score.

	*r*	*p*-Value
MFI-K		
General and Physical Fatigue	0.802	<0.001
Mental Fatigue	0.795	<0.001
Reduced Activity	0.716	<0.001
Motivation	0.759	<0.001
		<0.001
ESS	0.260	<0.001
HADS		
Anxiety	0.529	<0.001
Depression	0.630	<0.001

*p*-values were obtained by Pearson’s correlation analysis.

**Table 5 jcm-10-01588-t005:** Multiple regression analysis between MFI-K and other variables associated with fatigue.

	β	*p*-Value
Reflux Esophagitis (no or yes)	−0.358	0.700
GERD Symptoms (no or yes)	2.263	0.021
Age	0.014	0.705
Sex	2.488	0.001
HADS		
Anxiety	0.777	<0.001
Depression	1.617	<0.001

## Data Availability

The datasets generated or analyzed during the current study are available from the corresponding author on reasonable request.

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
