# Peer review of "Reflux Esophagitis and Fatigue: Are They Related?"

_jcm, 2021, doi:10.3390/jcm10081588_

Round 1

Reviewer 1 Report

-This is a prospective study in Korea evaluating association between GERD and fatigue using various PRO measures. N of 497. 

Major concerns:

1) It appears that the authors use various PRO measures that assess fatigue/anxiety/depression, but didn't use a PRO measure for GERD (to assess reflux symptom burden). This is a significant limitation as now a binary group is used (GERD, yes or no). Similar to fatigue/anxiety/depression, reflux symptoms have severity scale and with the lack of PRO measure, correlations are highly limited. 

2) I am not sure why table 1 includes lab values of WBC/AST/ALT/GRP/Creatinine, blood pressure values/fasting blood glucose, LDL, TC. The methods section or aims do not mention looking at lab differences. If above are included in the paper, authors should include it in methods/aims. The tables should be designed to include information that supports the primary aim of the paper. 

3) Authors did not define the comparison groups in the methods section. Table 3 notes difference between Asymptomatic erosive esophagitis and NERD (non-erosive reflux disease). However, NERD is defined by presence of symptoms AND abnormal esophageal acid exposure (which would need to be confirmed using esophageal pH testing). If pH testing was not done, this group can't be classified as NERD (and would just be eRE-, symptoms +). 

4) The authors graded esophagitis using LA classification per the methods, but the information about how many had LA A/B/C/D is not available anywhere in the results sections or the tables. There is brief mention about LA C/D in the limitations section of the discussion.

Minor concerns:

-There are some grammatical errors throughout the manuscript that would need to be corrected. 

Author Response

Reviewer: It appears that the authors use various PRO measures that assess fatigue/anxiety/depression, but didn't use a PRO measure for GERD (to assess reflux symptom burden). This is a significant limitation as now a binary group is used (GERD, yes or no). Similar to fatigue/anxiety/depression, reflux symptoms have severity scale and with the lack of PRO measure, correlations are highly limited. 

Author's response: Thank you for your important comment. We fully agree with the reviewer’s comment. Unfortunately, we did not use a PRO measure to assess reflux symptom burden. However, there were few patients with severe reflux esophagitis (LA-C; n=12, LA-D; n=1), so their symptoms were not severe.

We have added this point for limitations (page 15, line 5 ~ line 7) in the discussion

Reviewer: I am not sure why table 1 includes lab values of WBC/AST/ALT/GRP/Creatinine, blood pressure values/fasting blood glucose, LDL, TC. The methods section or aims do not mention looking at lab differences. If above are included in the paper, authors should include it in methods/aims. The tables should be designed to include information that supports the primary aim of the paper. 

Author's response: As the reviewer pointed out, we removed the variables (Lab values, blood pressure, etc) and revised Table 1.

Reviewer: Authors did not define the comparison groups in the methods section. Table 3 notes difference between Asymptomatic erosive esophagitis and NERD (non-erosive reflux disease). However, NERD is defined by presence of symptoms AND abnormal esophageal acid exposure (which would need to be confirmed using esophageal pH testing). If pH testing was not done, this group can't be classified as NERD (and would just be eRE-, symptoms +). 

Author's response: Thank you for your invaluable comments. We fully agree with the reviewer’s opinion. It is important to confirm NERD by 24hr pH-monitoring, but the presence of esophagitis was determined by endoscopic findings in this study.

Therefore, we modified the manuscript and table 3 using ENRD (endoscopy-negative reflux disease) instead of NERD (non-erosive reflux disease).

Reviewer: The authors graded esophagitis using LA classification per the methods, but the information about how many had LA A/B/C/D is not available anywhere in the results sections or the tables. There is brief mention about LA C/D in the limitations section of the discussion.

 Author's response: As the reviewer pointed out, we have added the information about how many had LA A/B/C/D in the result. à ( page 10, line 8 ~9)

Reviewer: There are some grammatical errors throughout the manuscript that would need to be corrected. 

The English in this document has been checked by at least two professional editors, both native speakers of English. For a certificate, please see:

http://www.textcheck.com/certificate/7XzMIM

Reviewer 2 Report

The paper entitled: Reflux Esophagitis and Fatigue: Are They Related? by the authors Sung-Goo Kang , Hyun jee Hwang , Youngwoo Kim , Junseak Lee , Jung Hwan Oh , Jinsu Kim , Chul-Hyun Lim , Seung Bae Youn , Sung Hoon Jung, deals with  the relative impact of GERD on fatigue compared to
other variables (age, sex and psychological disorders). The authors also 
investigated whether endoscopy-proved esophagitis or GERD symptoms were associated with fatigue. Altogether 497 eligible cases were analyzed. The authors conclude that fatigue was associated with GERD symptoms, women, anxiety and depression but
there was no significant association in which reflux esophagitis could cause
daytime sleepiness, fatigue, anxiety, or depression. Therefore, psychological factors are likely to cause fatigue and daytime sleepiness, rather than organic factors related to acid regurgitation. This in an interesting article with interesting results, and I am of that opinion that these results are well in sound with the clinical experience. It is worth of publishing this paper in JCM. The paper is very clearly presented, and it could be accepted for publication as such after minor evaluation in the quality of the English language.

Author Response

Reviewer: The paper entitled: Reflux Esophagitis and Fatigue: Are They Related? by the authors Sung-Goo Kang , Hyun jee Hwang , Youngwoo Kim , Junseak Lee , Jung Hwan Oh , Jinsu Kim , Chul-Hyun Lim , Seung Bae Youn , Sung Hoon Jung, deals with  the relative impact of GERD on fatigue compared to
other variables (age, sex and psychological disorders). The authors also 
investigated whether endoscopy-proved esophagitis or GERD symptoms were associated with fatigue. Altogether 497 eligible cases were analyzed. The authors conclude that fatigue was associated with GERD symptoms, women, anxiety and depression but
there was no significant association in which reflux esophagitis could cause
daytime sleepiness, fatigue, anxiety, or depression. Therefore, psychological factors are likely to cause fatigue and daytime sleepiness, rather than organic factors related to acid regurgitation. This in an interesting article with interesting results, and I am of that opinion that these results are well in sound with the clinical experience. It is worth of publishing this paper in JCM. The paper is very clearly presented, and it could be accepted for publication as such after minor evaluation in the quality of the English language. 

Author's response: We sincerely appreciate your meticulous comments on our research.

The English in this document has been checked by at least two professional editors, both native speakers of English. For a certificate, please see:

http://www.textcheck.com/certificate/7XzMIM

Round 2

Reviewer 1 Report

The authors have addressed my queries appropriately.